# Competition between H_4_PteGlu and H_2_PtePAS Confers *para*-Aminosalicylic Acid Resistance in *Mycobacterium tuberculosis*

**DOI:** 10.3390/antibiotics13010013

**Published:** 2023-12-21

**Authors:** Ji-Fang Yu, Jin-Tian Xu, Ao Feng, Bao-Ling Qi, Jing Gu, Jiao-Yu Deng, Xian-En Zhang

**Affiliations:** 1Faculty of Synthetic Biology, Shenzhen Institute of Advanced Technology, Chinese Academy of Sciences, Shenzhen 518055, China; 2Wuhan Institute of Virology, Center for Biosafety Mega-Science, Chinese Academy of Sciences, Wuhan 430071, China; 3University of Chinese Academy of Sciences, Beijing 100049, China; 4Shanghai Metabolome Institute-Wuhan (SMI), Wuhan 430000, China; 5National Laboratory of Biomacromolecules, Institute of Biophysics, Chinese Academy of Sciences, Beijing 100101, China

**Keywords:** *Mycobacterium tuberculosis*, *para*-aminosalicylic acid, tetrahydrofolate, *thyA*

## Abstract

Tuberculosis remains a serious challenge to human health worldwide. *para*-Aminosalicylic acid (PAS) is an important anti-tuberculosis drug, which requires sequential activation by *Mycobacterium tuberculosis* (*M. tuberculosis*) dihydropteroate synthase and dihydrofolate synthase (DHFS, FolC). Previous studies showed that loss of function mutations of a thymidylate synthase coding gene *thyA* caused PAS resistance in *M. tuberculosis*, but the mechanism is unclear. Here we showed that deleting *thyA* in *M. tuberculosis* resulted in increased content of tetrahydrofolate (H_4_PteGlu) in bacterial cells as they rely on the other thymidylate synthase ThyX to synthesize thymidylate, which produces H_4_PteGlu during the process. Subsequently, data of in vitro enzymatic activity experiments showed that H_4_PteGlu hinders PAS activation by competing with hydroxy dihydropteroate (H_2_PtePAS) for FolC catalysis. Meanwhile, over-expressing *folC* in Δ*thyA* strain and a PAS resistant clinical isolate with known *thyA* mutation partially restored PAS sensitivity, which relieved the competition between H_4_PteGlu and H_2_PtePAS. Thus, loss of function mutations in *thyA* led to increased H_4_PteGlu content in bacterial cells, which competed with H_2_PtePAS for catalysis by FolC and hence hindered the activation of PAS, leading to decreased production of hydroxyl dihydrofolate (H_2_PtePAS-Glu) and finally caused PAS resistance. On the other hand, functional deficiency of *thyA* in *M. tuberculosis* pushes the bacterium switch to an unidentified dihydrofolate reductase for H_4_PteGlu biosynthesis, which might also contribute to the PAS resistance phenotype. Our study revealed how *thyA* mutations confer PAS resistance in *M. tuberculosis* and provided new insights into studies on the folate metabolism of the bacterium.

## 1. Introduction

Tuberculosis (TB), caused by *M. tuberculosis*, is an ancient infectious disease. Recent data released by the World Health Organization show that around 10 million people fell in with the disease every year worldwide [1]. The increasing spread of drug-resistant *M. tuberculosis* makes TB treatment more difficult, and drug resistance has become one of the major challenges. The best way to solve the above problem is to introduce new anti-TB drugs. However, no new first-line drug has been introduced in clinical TB treatment for more than 50 years, since rifampicin [2]. Therefore, rational use of existing anti-tuberculosis drugs is necessary. In addition, researchers also have made efforts in using phages as an individual or supplementary therapy to treat *M. tuberculosis* infections [3].

Folate is an essential nutrient for all sorts of life. Bacteria need to synthesize folate de novo, but mammals are unable to synthesize it, which makes the bacterial de novo folate biosynthesis pathway an ideal target for developing new antibacterial drugs [4]. As is well known, dihydropteroate (H_2_Pte) is synthesized by dihydropteroate synthetase (DHPS, FolP) using *para*-aminobenzoic acid (*p*ABA) and 7,8-dihydropterin pyrophosphate (H_2_PtePP) as substrates, which is further converted into dihydrofolate (H_2_PteGlu) by FolC (Figure 1) [5]. Dihydrofolate reductase (DHFR, DfrA or RibD) and thymidylate synthase (ThyA or ThyX) maintain the interconversion and balance between H_2_PteGlu, H_4_PteGlu and 5, 10-methylenetetrahydrofolate (5, 10-m-H_4_PteGlu) (Figure 1). PAS was first used as a first-line anti-TB drug in 1946 [6], and is presently still used for treating multiple drug-resistant TB [7]. The mechanism of action of PAS had been gradually discovered over 70 years of clinical utilization. As a structural analogue of *p*ABA, PAS is firstly catalyzed by the FolP1 of *M. tuberculosis* to form H_2_PtePAS, an analogue of H_2_Pte. Subsequently, H_2_PtePAS was further catalyzed by the FolC, yielding H_2_PtePAS-Glu [5] (Figure 1). Ultimately, H_2_PtePAS-Glu inhibited the activity of *M. tuberculosis* DfrA (Figure 1), resulting in bacterial growth inhibition and cell death [8].

Although the mechanism of PAS action has been elucidated, its mechanisms of resistance still await investigation. Until the present, confirmed molecular markers associated with PAS resistance in *M. tuberculosis* clinical isolates included mutations of *folC* [9,10,11], *thyA* [9,11,12,13], and *ribD* [8,9,11]. Among them, *folC* or *thyA* gene mutations were the main reasons for PAS resistance, accounting for two-thirds of the PAS resistant clinical isolates [9,11,14]. Molecular mechanisms of PAS resistance caused by *folC* and *ribD* mutations have been elucidated [8,10]. Our previous research showed that H_2_Pte binding pocket variants of FolC failed to activate H_2_PtePAS to H_2_PtePAS-Glu, hindering the activation of PAS and hence conferring resistance to PAS [10]. On the other hand, *ribD* could serve as an alternative for DHFR, as mutations in the promoter region of the gene could cause over-expression of *ribD*, and thus lead to PAS resistance [8]. However, the molecular mechanism of PAS resistance caused by *thyA* mutations still remains unclear, though the association between *thyA* mutations and PAS resistance has been established for nearly two decades [13]. According to the data of epidemiological analysis, *thyA* mutations were identified in about 1/3 of the PAS resistant *M. tuberculosis* clinical isolates [9,11,12]. Thus, unravelling the mechanism of PAS resistance caused by *thyA* mutations will broaden our understanding of folate metabolism in *M. tuberculosis* and be useful for guiding the clinical administration of PAS. To elucidate how *thyA* mutations caused PAS resistance in *M. tuberculosis*, the *thyA* gene was deleted in H37Ra using the phage-mediated allelic exchange method, and a clinical PAS resistant isolate F461 harboring the *thyA* R235P mutation was selected [14]. Subsequently, the effect of *thyA* deletion on bacterial H_4_PteGlu content was determined by UPLC-MS/MS. Then, the competition for catalysis of FolC between H_4_PteGlu and H_2_PtePAS was analyzed by in vitro enzymatic activity assays. Meanwhile, *folC* was over-expressed in the *thyA* deletion mutant and the selected PAS resistant clinical isolate, PAS susceptibilities of these two strains were tested. The level of FolC in ThyA deficiency strain was explored by RNA-seq and Western blot assays. The results are presented herein.

## 2. Results

### 2.1. thyA Deletion Leads to High Level PAS Resistance in M. tuberculosis

Considering the genetic complexity of clinical isolates, and also high similarity of mechanisms of PAS action and resistance between H37Ra and H37Rv [10], we constructed the *thyA* deletion strain in H37Ra to elucidate the molecular mechanism of how *thyA* mutations lead to PAS resistance in *M. tuberculosis*. H37Ra Δ*thyA* showed a significant growth defect (Appendix A), which is consistent with the observation in H37Rv Δ*thyA* [15]. Subsequently, the susceptibility to PAS was determined. The results showed that *thyA* deletion led to a hundreds of times increase in minimum inhibitory concentration (MIC) of PAS to *M. tuberculosis* (Table 1), which is consistent with clinical data [13]. After that, recombinant plasmids carrying *thyA* or *thyX* genes from *M. tuberculosis* H37Ra were used to transform H37Ra and H37Ra Δ*thyA*, respectively. Plasmid-borne expression of *thyA* restored PAS sensitivity of the *thyA* deletion strain, but that of *thyX* could not (Table 1). We noticed that over-expression of *thyA* and *thyX* both caused an eight times increase in PAS MIC (Table 1).

### 2.2. folC Over-Expression Partially Restores PAS Sensitivity in thyA Functional Deficient Strains

Previous researches have confirmed that blocking the incorporation of PAS into folate synthesis pathway leads to high level resistance to PAS in *M. tuberculosis* [8,10]. To assess whether the high-level resistance to PAS of the *thyA* deletion strain was related to the efficiency of PAS incorporation, core genes *folP1*, *folC*, and *dfrA* of the folate biosynthesis pathway were over-expressed in H37Ra and H37Ra Δ*thyA*. The results showed that plasmid-borne expression of *folP1* and *folC* in H37Ra led to increased sensitivity to PAS, as demonstrated by the reduced MICs (four times for *folP1* over-expression and two times for *folC* over-expression) (Table 2). As the target for bio-activated PAS, *dfrA* over-expression increased the PAS MIC by thousands of times (Table 2). Over-expression of *folP1* in H37Ra Δ*thyA* also led to a four-times decrease in PAS MIC, which was consistent with that in H37Ra (Table 2). However, over-expressing *folC* in H37Ra Δ*thyA* led to a 16-times decrease in PAS MIC, and over-expressing *dfrA* in H37Ra Δ*thyA* did not change the PAS MIC (Table 2). To further prove that over-expressing *folC* could reverse the high-level PAS resistance phenotype in *thyA* functional deficient strains, *folC* was over-expressed in the PAS resistant clinical isolate harboring the *thyA* R235P mutation. As shown in Table 2, *folC* over-expression also led to a 10-times decrease in PAS MIC in the clinical isolate.

### 2.3. The Expression Level of folC Gene and FolC Protein Remain Unchanged in H37Ra ΔthyA

To further explore the role of *folC* in PAS resistance caused by ThyA functional deficiency, we detected the expression level of *folC* in wild-type and *thyA* deletion strain. Western blot assay was performed to compare the expression level of FolC between wild-type and *thyA* deletion strain, and the results showed that the FolC expression level was not significantly changed in the *thyA* deletion strain (Figure 2A,B). Meanwhile, RNA-seq data also showed that the expression level of *folC* was not significantly changed in the *thyA* deletion strain (Figure 2C).

### 2.4. thyA Deletion Leads to Increased H_4_PteGlu Content in Bacterial Cells

There are two types of thymidylate synthase, ThyA and ThyX, in *M. tuberculosis* [15], and the thymidylate synthase function is mainly performed by ThyA. ThyA uses 5, 10-m-H_4_PteGlu as methyl donor to generate H_2_PteGlu and maintain the balance of folate metabolism (Figure 1) [15,16], and ThyX uses 5, 10-m-H_4_PteGlu as methyl donor to generate H_4_PteGlu (Figure 1) [16,17]. After the loss of ThyA function, the bacterium relies on ThyX for synthesizing thymidylate [15]. Thus, we speculated that the H_4_PteGlu content would increase in ThyA deficient strains. As expected, we observed an obvious increase in H_4_PteGlu content in the *thyA* deletion strain compared to the wild-type strain (Figure 3).

### 2.5. Comparison of Catalytic Efficiency of FolC on H_2_Pte, H_4_PteGlu and H_2_PtePAS

FolC was demonstrated to be a bifunctional enzyme in *Escherichia coli* (*E. coli*) which not only converted H_2_Pte into H_2_PteGlu, but also added glutamic acid tail to H_4_PteGlu [18,19]. Therefore, we speculated that H_4_PteGlu would also compete with H_2_PtePAS for catalysis activity of FolC in *M. tuberculosis*, thus hindering the activation process of PAS. To test this speculation, catalytic efficiency of FolC on H_2_Pte, H_4_PteGlu, and H_2_PtePAS was compared. The results showed that, under the same reaction conditions, FolC could convert about 85% H_2_Pte and 50% H_4_PteGlu, but only about 12% H_2_PtePAS (Figure 4).

### 2.6. H_4_PteGlu Hinders the Activation of PAS by FolC

To further demonstrate whether H_4_PteGlu could hinder the activation of PAS by FolC, H_2_PtePAS was synthesized by purified recombinant FolP1 using H_2_PtePP and PAS as substrates [10]. H_2_PtePAS was analyzed by UPLC-MS/MS (Figure 5A, Appendix A). FolC catalytic activity was analyzed using H_2_PtePAS instead of H_2_Pte as a substrate. Consistent with previous reports [5,10], FolC could catalyze the ligation of L-glutamic acid to H_2_PtePAS generating H_2_PtePAS-Glu, which was confirmed by HPLC-MS/MS (Figure 5B, Appendix A). We then sought to understand the effect of H_4_PteGlu on H_2_PtePAS activation by FolC, and different concentrations (10 µM and 50 µM) of H_4_PteGlu were added into the FolC reaction mixture using H_2_PtePAS as substrate. As shown in Figure 5C, when H_4_PteGlu was added into the reaction mixture, the catalytic efficiency of FolC for H_2_PtePAS decreased remarkably.

## 3. Discussion

Folates, especially derivatives of H_4_PteGlu, are one carbon carriers required by the biosynthesis of purines, thymidylate, methionine, serine, and glycine, thus making them essential for all sorts of lives [20,21]. Bacteria must synthesize these essential cofactors *de novo*, while mammal can intake them from their diet [4]. This difference makes the bacterial *de novo* folate biosynthesis pathway an ideal target for developing new antibacterial drugs [4]. Although thousands of folates antagonists have been designed for folate biosynthesis pathway heretofore, PAS is the only one used for TB treatment with a unique mode of action only observed in *M. tuberculosis* complex. Thus, better understanding the mechanisms of PAS resistance in *M. tuberculosis* will benefit the development of new antifolates against this bacterium.

As the first molecular marker for PAS resistance in *M. tuberculosis* clinical isolates, *thyA* gene mutations have been identified for nearly two decades [13], but the molecular mechanism of how these mutations lead to PAS resistance remains unknown. Ten years later, when probing the molecular mechanism of PAS resistance caused by *folC* mutation [10], we noticed that though FolC could also catalyze the conversion of H_2_PtePAS to H_2_PtePAS-Glu, but the catalytic efficiency was much lower than that of the natural substrate H_2_Pte, implying that the bio-activation process of PAS might be vulnerable to interference of natural metabolite of folate biosynthesis. Indeed, exogenous H_2_Pte made *M. tuberculosis* more resistant to PAS [10]. Previous studies have showed that FolC could not only convert H_2_Pte into H_2_PteGlu, but also add glutamic acid tail to H_4_PteGlu in *E. coli* [18,19]. In this study, we found that *M. tuberculosis* FolC is also bifunctional. In addition, its catalytic efficiency for H_2_PtePAS is remarkably lower than that for H_4_PteGlu (Figure 4), implying intracellular H_4_PteGlu may interfere the activation of PAS by FolC. As expected, the in vitro biochemical experiments showed that H_4_PteGlu hinders the conversion of H_2_PtePAS to H_2_PtePAS-Glu in a concentration-dependent manner (Figure 5C). Since *M. tuberculosis* is not able to intake exogenous H_4_PteGlu, it is not possible to test the effect of exogenous H_4_PteGlu on PAS susceptibility. Alternatively, we compared the H_4_PteGlu content between H37Ra and the *thyA* deletion mutant, and found that the H_4_PteGlu content in the *thyA* deletion mutant was significantly higher than that of the wild-type strain (Figure 3). This is not surprising since the bacterium has to solely rely on ThyX to synthesize thymidylate in the absence of ThyA, and utilization of the former yields H_4_PteGlu. Since the expression level of FolC remained unchanged in the *thyA* deletion mutant, increased H_4_PteGlu content could hinder the conversion of H_2_PtePAS since they compete for the same protein. Correspondingly, this competition could be mitigated by over-expression of the target protein FolC. As expected, over-expression of *folC* could reverse the PAS resistance phenotype caused by *thyA* deletion or clinical *thyA* R235P mutation (Table 2). We noticed that the PAS resistance phenotype caused by *thyA* deletion or mutation could only be partially restored by *folC* over-expression, suggesting the existence of other mechanisms for PAS resistance caused by functional deficiency of ThyA.

When assessing whether the resistance to PAS of the *thyA* deletion mutant was related to the efficiency of PAS activation, we over-expressed *folP1*, *folC*, and *dfrA* in H37Ra and H37Ra Δ*thyA*. To our surprise, over-expression of *dfrA* in the *thyA* deletion mutant did not affect the susceptibility to PAS (Table 2), suggesting either the lack of DfrA protein or loss of function of DfrA in the *thyA* deletion mutant. Previous works also showed that *thyA* and *dfrA* double deletion mutants had been identified in *M. tuberculosis* clinical isolates from different countries [11,22]. Thus, in the absence of *thyA*, *M. tuberculosis* discards the commonly used DHFR Rv2763c (DfrA), and switches to another alternative to synthesize H_4_PteGlu. Although RibD was shown to be an alternative DHFR in *M. tuberculosis*, previous research revealed that RibD could only replace DfrA when it was highly over-expressed in a multi-copy plasmid [8], suggesting that the dihydrofolate reductase activity of RibD is quite low, which was confirmed by subsequent biochemical analysis [23]. Zheng et al. found that mutations in the promoter region of *ribD* could cause over-expression of *ribD* [8]. To determine whether RibD is the alternative DHFR in the absence of ThyA in *M. tuberculosis*, we further analyzed genome sequences of isolates with frameshift or deletion mutations in *thyA* or *dfrA* genes from previous studies and NCBI database. The results showed that there was no mutation in either the promoter region (300 bp upstream start codon) or the coding sequence (CDS) of the *ribD* gene in ThyA or DfrA deficient clinical isolates (Appendix A). Moreover, our RNA-seq data also showed that the expression level of *ribD* remained unchanged in the *thyA* deletion mutant (Appendix A). Therefore, RibD is not the alternative DHFR in the absence of ThyA. What the alternative DHFR is in the absence of ThyA requires further investigation. It will be important to test if the alternative DHFR would be more resistant to the inhibition of H_2_PtePAS-Glu, since over-expressing *folC* could only partially restore PAS sensitivity to the *thyA* deletion mutant.

Previous studies already showed that the C^−16^T mutation in the upstream regulatory region of *thyX* could lead to increased expression of *thyX* and PAS resistance in *M. tuberculosis* [24,25]. Thus, it is not surprising to see that over-expressing *thyX* led to PAS resistance in H37Ra. The fact that over-expressing *thyX* in the *thyA* deletion mutant did not affect PAS susceptibility of the latter indicated that over-expressing *thyX* and deleting *thyA* in *Mtb* might share the same mechanism of PAS resistance. In addition, over-expressing *thyA* in H37Ra also led to low level PAS resistance. Considering the role of ThyA in folate salvage, we speculated that the intracellular H_2_PteGlu content might be increased when over-expressing *thyA*; this would in turn reduce the demand for dihydrofolate biosynthesis through FolC. Previous studies showed that FolC was critical for the bio-activation of PAS, and decreased FolC enzymatic activity caused PAS resistance [8,10].

In conclusion, our results showed that functional deficiency of ThyA led to increased H_4_PteGlu content of the bacterial cells, which competed with H_2_PtePAS for FolC catalysis, thus hindered the activation of PAS and conferred PAS resistance in *M. tuberculosis*. Meanwhile, our study also suggested that *M. tuberculosis* could switch from Rv2763c to a yet unknown alternative DHFR in the absence of *thyA*, and further investigation is required to identify the protein and elucidate its role on PAS resistance caused by ThyA functional deficiency. Our study broadens the understanding of folate metabolism in *M. tuberculosis* and might be useful for guiding the clinical administration of PAS.

## 4. Materials and Methods

### 4.1. Bacterial Strains, Plasmids, and Growth Conditions

Clinical isolate F461, *M. tuberculosis* H37Ra and its derivative strains were cultured at 37 °C in 7H9 broth (Difco, St. Louis, MO, USA) supplemented with 10% (*v*/*v*) oleic acid-albumin-dextrose-catalase (OADC, Difco), 0.5% (*v*/*v*) glycerol, and 0.05% (*v*/*v*) Tween 80 (Sigma-Aldrich, St. Louis, MO, USA), or on 7H10 agar medium (Difco) supplemented with 10% (*v*/*v*) OADC and 0.5% (*v*/*v*) glycerol. *Mycobacterium smegmatis* mc^2^155 was grown in Middlebrook 7H9 medium or 7H10 agar medium. *E. coli* strains HB101 and BL21 (DE3) were cultured in Luria-Bertani (LB) medium (Difco), or on LB agar plates at 37 °C. Plasmids pMAL-c2X (New England BioLabs, Beverly, MA, USA), pET-28a (Novagen, Madison, WI, USA), and pMV261 were used for the construction of expression plasmids. All bacteria strains, plasmids, and primers used in this study are described in detail in Appendix A.

### 4.2. Antibiotics and Chemicals

These concentrations of antibiotics (75 μg mL^−1^ and 150 μg mL^−1^ hygromycin (Sigma-Aldrich), 25 μg mL^−1^ and 100 μg mL^−1^ Kanamycin (MD Bio, Inc., Qingdao, China), and 150 μg mL^−1^ ampicillin (MD Bio, Inc.)) were used to culture bacteria, unless otherwise indicated. H_2_Pte and H_4_PteGlu were purchased from Schircks Laboratories. PAS (Sigma-Aldrich) was used at indicated concentrations.

### 4.3. Genetic Manipulation of Mycobacterial Strains

*folC*, *thyA*, *thyX*, *dfrA*, and *folP1* were amplified from wild-type *M. tuberculosis* H37Ra genomic DNA using PCR with the primers (Appendix A). The purified amplicon was digested and ligated to pMV261, generating pMV261-*folC*, pMV261-*thyA*, pMV261-*dfrA*, pMV261-*folP1*, and pMV261-*thyX*. *M. tuberculosis* strain was transformed with sequence-confirmed pMV261 recombinant plasmid, then plated on 7H10 medium containing 25 μg mL^−1^ kanamycin. After 3 weeks of incubation at 37 °C, single colonies were purified and liquid cultures were prepared for the extraction of genomic DNA and determination of PAS MICs, separately. The presence of pMV261 recombinant plasmid was verified by PCR amplification using primers specific for pMV261-JDFP and pMV261-JDRP (Appendix A).

A modified strategy for phage-mediated allelic exchange [26] was used to construct *M. tuberculosis* H37Ra Δ*thyA* mutant. Briefly, the native copy of *thyA* was deleted by specialized transduction using phAE159 containing a hygromycin resistance cassette. All primers used are listed in Appendix A.

### 4.4. Purification of Recombinant FolP1 and FolC

FolP1 and FolC proteins were purified as previously reported [10]. Briefly, *folP1* and *folC* were amplified from *M. tuberculosis* H37Ra genomic DNA using specific primers (Appendix A) and separately cloned into pET28a to yield pET28a::*folP1* to introduce an N-terminal hexa-histidine tag and into pMAL-c2X to yield pMAL-c2X::*folC* to introduce an N-terminal maltose-binding protein (MBP) tag linked with a factor Xa cleavage site. The sequence-confirmed recombinant plasmids were transformed into *E. coli* BL21 (DE3). The cells were grown at 37 °C in LB broth containing 150 μg mL^−1^ ampicillin or 100 μg mL^−1^ kanamycin to an OD_600_ of ~0.6. Isopropyl-β-D-thiogalactopyranoside (IPTG, Acmec, China) was added to 0.25 mM, then the cells were incubated further at 16 °C for 20 h. The bacterial cells were harvested by centrifugation, disrupted by sonication, and clarified by centrifugation.

Recombinant FolP1 protein was purified over prewashed nickel–nitrilotriacetic acid HisTrap HP affinity resin (GE Healthcare, Little Chalfont, Buckinghamshire, UK). Nonspecifically bound protein was removed by washing the resin with 50 mM Tris-HCl, 0.5 M NaCl, and 60 mM imidazole (pH 8.0). Recombinant FolP1 was eluted with 50 mM Tris-HCl, 0.5 M NaCl, and 400 mM imidazole (pH 8.0), and analyzed by SDS-PAGE.

Recombinant FolC proteins were first purified over an amylose resin column (New England BioLabs). The FolC protein obtained from the first purification contains MBP tag. To remove the MBP tag, the purified samples were incubated with factor Xa at 4 °C overnight in reaction buffer (20 mM HEPES (pH 8.0), 100 mM NaCl, 2 mM CaCl_2_, and 10% glycerol). Then, the cleavage mixtures were dialyzed against 50 mM phosphate buffer (pH 8.0). The samples were loaded on a HiTrap DEAE FF column (GE Healthcare), and a step gradient from 50 mM to 1 M NaCl in phosphate buffer was applied to elute FolC. The fractions were then analyzed by SDS-PAGE. Recombinant FolC was eluted with 300 mM NaCl.

### 4.5. Western Blot Assay

The H37Ra and H37Ra Δ*thyA* strains were cultured at 37 °C in 10 mL of 7H9 medium and harvested at logarithmic phase by centrifugation. For Western blot analysis, bacterial cells were resuspended in phosphate buffer saline (PBS, pH 7.0), then lysed using zirconium beads. Protein samples acquired from the supernatant after centrifugation. The protein concentration of the supernatant was determined using the NanoDrop2000 (Thermo, Waltham, MA, USA). Then, the protein samples were separated by SDS-PAGE and immediately transferred to a polyvinylidene difluoride membrane (Merck Millipore, Darmstadt, Germany) by a Bio-Rad SD device (Bio-Rad Laboratories, Hercules, CA, USA) at 15 V for 30 min. Finally, the proteins were probed with rabbit FolC polyclonal antibody (ABclonal biotechnology, Wuhan, China, Cat. No. WG-00133D).

### 4.6. RNA-Seq Analysis

Mycobacterial strains were grown in 7H9 to mid logarithmic phase and were collected by centrifugation. Total RNA was extracted using RNeasy mini kit (Qiagen, Hilden, Germany). Library constructions were prepared using TruSeq Stranded Total RNA Sample Preparation kit (Illumina, San Diego, CA, USA), and RNA sequencing was conducted on Illumina NovaSeq6000 at Beijing Novogene Corporation. The insert size conformation of purified libraries was validated by an Agilent 2100 bioanalyzer (Agilent Technologies, Santa Clara, CA, USA). Bowtie2 was used to map the cleaned reads to the M. tuberculosis H37Ra genome acquired from the National Center for Biotechnology Information (NCBI) (https://www.ncbi.nlm.nih.gov/nuccore/CP000611.1) (Accessed on 25 May 2023). Then, HTSeqv0.6.1 was run with a reference annotation to generate fragments per kilobase of exon model per million mapped reads values for estimation of fold changes. Three biological replicates were used in RNA-seq and the *p*- and q-values were calculated. The differentially expressed genes were selected using the following filter criteria: q-value < 0.005 and |log2 (fold change)| > 1. Raw RNA sequencing data have been deposited at NCBI Sequence Read Archive, Accession PRJNA1005084.

### 4.7. In Vitro Enzymatic Activity Assays

The dihydrofolate synthase activities of FolC using H_2_Pte, H_2_PtePAS, and H_4_PteGlu as substrates were measured and H_2_PtePAS was enzymatically synthesized as previously described [5,10]. Briefly, the reaction mixture contained 1.2 µM FolP1, 40 mM Tris-20 mM glycine (pH 9.5), 5 mM MgCl_2_, 1 mM DTT, 200 mM NaCl, appropriate amounts of 6-hydroxymethyl-7,8-pterin pyrophosphate (H_2_PtePP), and 250 µM PAS. The reaction mixture was incubated at 37 °C until no increment of H_2_PtePAS accumulation was detected by UPLC-MS/MS. FolP1 was removed by passing through a 10-kDa Microcon centrifugal filter, and 325 µL of the remaining reaction mixture was used as a substrate for FolC. The FolC reaction mixture contained 0.5 µM FolC protein, 2.5 mM ATP, and 0.5 mM L-glutamate in 100 mM Tris-50 mM glycine (pH 9.5), 10 mM MgCl_2_, 5 mM DTT, 100 mM KCl, 50 mM NaCl, 10% glycerol, appropriate amounts H_2_PtePAS, and the presence/absence of H_4_PteGlu. The mixture was incubated at 37 °C for 15 min. H_2_Pte, H_2_PtePAS, and H_4_PteGlu were identified by UPLC-MS/MS. UPLC column was Waters ACQUITY UPLC HSS T3 Column (2.1 × 100 mm, 1.8 μm particles) using a flow rate of 0.4 mL/min at 40 °C during a 6 min gradient (0–1 min from 2% B to 1% B, 1–3.5 min from 1% B to 50% B, 3.5–3.8 min from 50% B to 95% B, 3.8–6 min 95% B), while using the solvents A (water containing 20 mM ammonium acetate) and B (methanol). Electrospray ionization was performed using the positive ion mode, the pressure of the nebulizer was 30 psi, the dry gas temperature was 325 °C with a flow rate of 11 L/min, the sheath gas temperature was 350 °C with a flow rate of 10 L/min, and the capillary was set at 4000 V. Multiple reaction monitoring (MRM) was used for the quantification of screening fragment ions. Peak determination and peak area integration were performed using Mass Hunter Workstation software (Agilent, Version B.08.00). *p*-values (*p*) were calculated using *t*-tests. The graphs for the transformation rate of H_2_Pte, H_4_PteGlu, and H_2_PtePAS were prepared using GraphPad Prism.

### 4.8. Drug Susceptibility Testing

Mycobacterial cells were cultured to mid-log phase (OD_600_: 0.5–1.0) and diluted to about 10^5^ cfu mL^−1^ using 10-fold serial dilutions in fresh 7H9 medium with or without 10% OADC. Then, bacterial cells were plated on 7H10 agar solid plates containing various concentrations of PAS (0, 0.00125, 0.0025, 0.005, 0.01, 0.02, 0.04, 0.08, 0.16, 0.32, 0.64, 1.28, 2.56, 5.12, 10.24, 20.48, 40.96, and 81.92 μg mL^−1^). PAS was purchased from Sigma-Aldrich and solubilized according to the manufacture’s recommendations. Plates were then incubated at 37 °C for 21 days. The MIC was defined as the lowest concentration of antibiotics required to inhibit 99% of CFUs after this culture period. The MICs were performed through two technical repetitions using three biological replicates. All of the bacteria strains used are listed in Appendix A.

### 4.9. Determination of H_4_PteGlu Content In Vivo

Bacteria samples (~5 × 10^9^ cfu) were re-suspended in 0.4 mL pre-cooled 20 mM HEPES (containing 2% vitamin C and 1% dithiothreitol, pH 7.0) and subjected to three liquid nitrogen freeze–thaw cycles and zirconia bead grinding before sonication in an ice bath for 15 cycles (1 min pulse followed by 1 min pause). The above extraction procedure was repeated three times. The mixture was then centrifuged for 10 min at 12,000× *g* at 4 °C, and each supernatant was filtered using a 0.22 µm membrane filter before UPLC-MS/MS analysis. The samples were detected as above with some changes. Briefly, the samples (5 μL) were individually injected on an UPLC column (Agilent ZORBAX Eclipse Plus C_18_ column, 2.1 × 100 mm, 1.8 μm particles) using a flow rate of 0.4 mL/min at 50 °C using the solvents A (water containing 0.1% (*v*/*v*) formic acid) and B (methanol containing 0.1% (*v*/*v*) formic acid). The bacterial biomasses of the individual samples were determined by colony counting method. All data obtained by metabolomics were averaged from the independent sextuplicates. *p*-values (*p*) were calculated using *t*-tests. The graphs for the determination of H_4_PteGlu in vivo were prepared using GraphPad Prism.

### 4.10. Comparative Analysis of Variants in M. tuberculosis Genomes

*M. tuberculosis* clinical isolates with complete or partial deletion of *thyA* or *dfrA* were extensively collected from previous studies [11,22,27] and the NCBI database (https://www.ncbi.nlm.nih.gov/genome/browse#!/prokaryotes/mycobacterium%20tuberculosis) (Accessed on 7 December 2022). A total of 31 *M. tuberculosis* genomes from clinical isolates were obtained, and the mutations in the promoter region (300 bp upstream start codon) or the CDS of *ribD* were analyzed in these isolates (Appendix A). All of the raw reads were available. The acquired reads were subjected to quality assessment using FastQC v.0.11.9. Subsequently, low-quality sequences were removed and trimmed using fastp. Reads shorter than 50 bp were discarded, the last 10 bp were trimmed, and bases with an average quality below 25 were removed using a sliding window of 20 bp. Finally, variant calling against the *M. tuberculosis* H37Rv (NC_000962.3) genome was performed using the Snippy pipeline.

### 4.11. Statistical Analysis

GraphPad Prism 8.0.1 was used to analyze all experimental data, adopting the two-tailed unpaired *t*-test method. Mean ± standard deviation (SD) was adopted to express the experimental data.

## Figures and Tables

**Figure 1 antibiotics-13-00013-f001:**
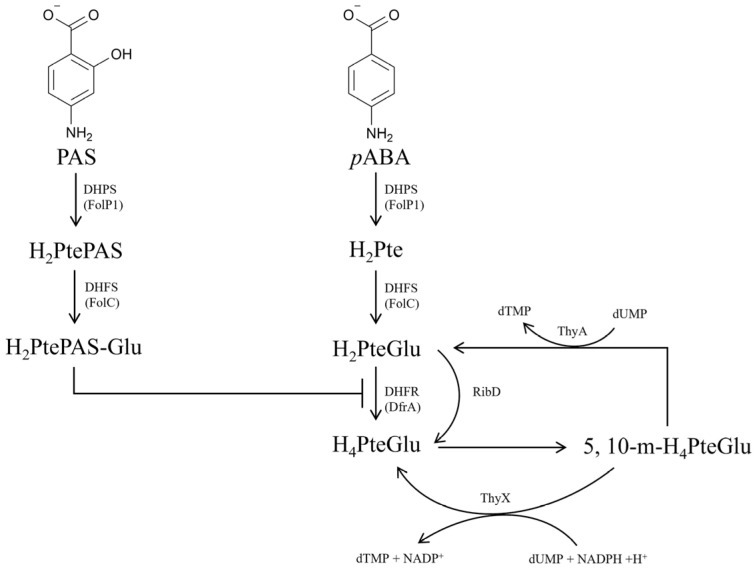
Schematic diagram of the mechanism of PAS action. PAS, *para*-aminosalicylic acid; *p*ABA, *para*-aminobenzoic acid; H_2_PtePAS, hydroxy dihydropteroate; H_2_Pte, dihydropteroate; H_2_PtePAS-Glu, hydroxy dihydrofolate; H_2_PteGlu, dihydrofolate; H_4_PteGlu, tetrahydrofolate; 5, 10-m-H_4_PteGlu, 5, 10-methylenetetrahydrofolate; DHPS/FolP1, dihydropteroate synthetase; DHFS/FolC, dihydrofolate synthase; DHFR/DfrA, dihydrofolate reductase; ThyA, thymidylate synthase; ThyX, thymidylate synthase; RibD, bifunctional diaminohydroxyphosphoribosylaminopyrimidine deaminase/5-amino-6-(5-phosphoribosylamino) uracil reductase.

**Figure 2 antibiotics-13-00013-f002:**
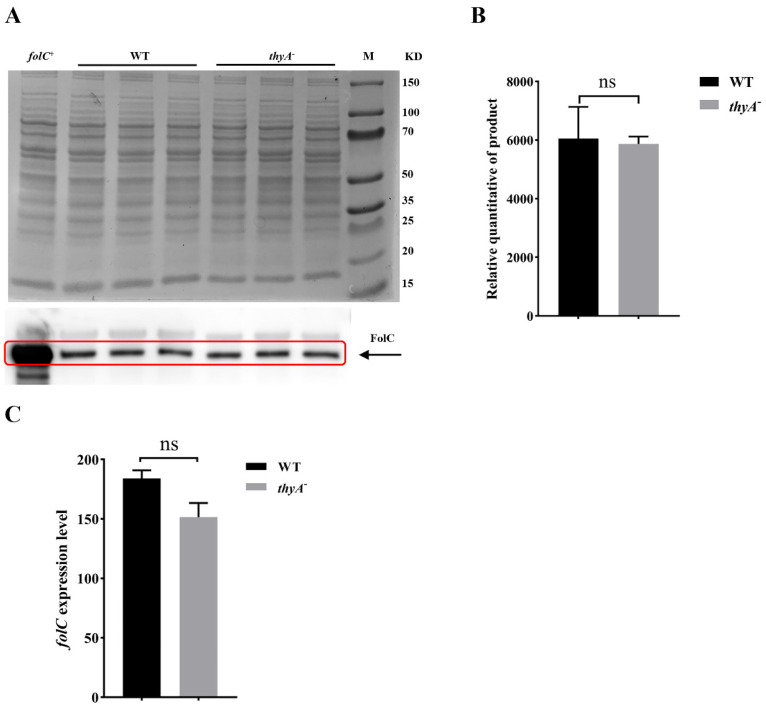
The expression of *folC* remains unchanged in ThyA functional deficient strain. (**A**) Comparison of the expressional level of FolC during the exponential phase in H37Ra (WT) and H37Ra Δ*thyA* (*thyA*^−^) by Western blot assay. Upper part: Total protein was normalized to 25 μg of each strain, then electrophoresed by SDS-PAGE and stained by Coomassie brilliant blue. Lower part: Western blot analysis of total protein immunoblotted with rabbit FolC polyclonal antibody. Experiments were repeated at least three times, and were performed three biological replicates each time. (**B**) Relative quantitative of FolC product by Western blot assay. ns, no significance. (**C**) Comparison of the transcriptional level of the gene *folC* during the exponential phase in H37Ra (WT) and H37Ra Δ*thyA* (*thyA*^−^) by RNA-seq. ns, no significance.

**Figure 3 antibiotics-13-00013-f003:**
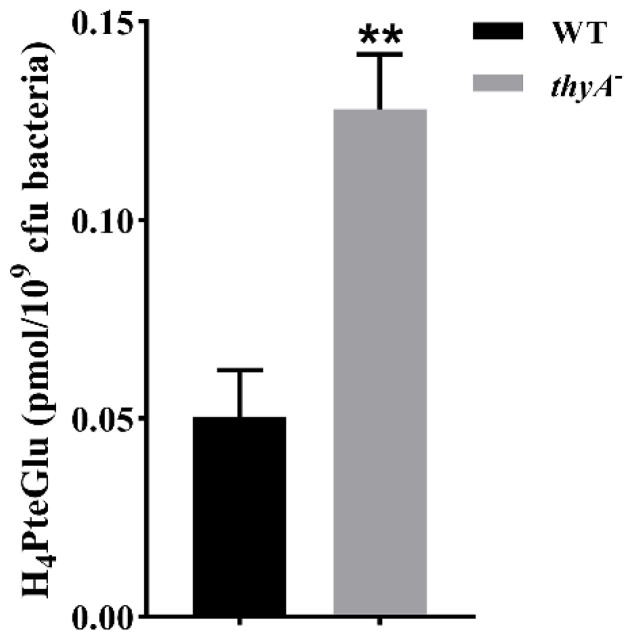
The quantitative detection of H_4_PteGlu by UPLC-MS/MS in *thyA* deletion strain. Cell-associated H_4_PteGlu was extracted from H37Ra (WT) and H37Ra ∆*thyA* (*thyA*^-^). The experiments were performed using six biological replicates. *p*-values (*p*) were calculated using *t*-tests. ** *p* < 0.01.

**Figure 4 antibiotics-13-00013-f004:**
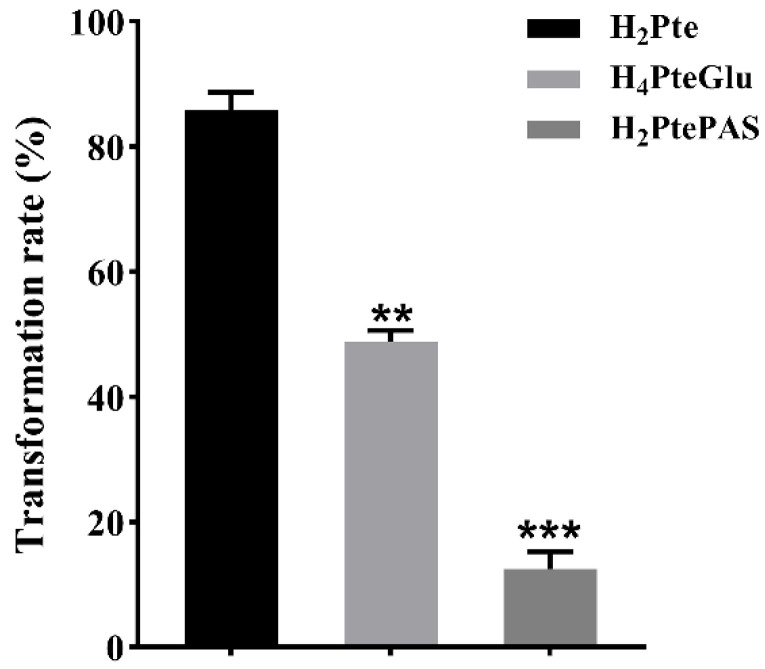
Catalytic utilization of H_2_Pte, H_4_PteGlu, and H_2_PtePAS by DHFS. The experiments were performed using three biological replicates. *p*-values (*p*) were calculated using *t*-tests. ** *p* < 0.01, *** *p* < 0.001.

**Figure 5 antibiotics-13-00013-f005:**
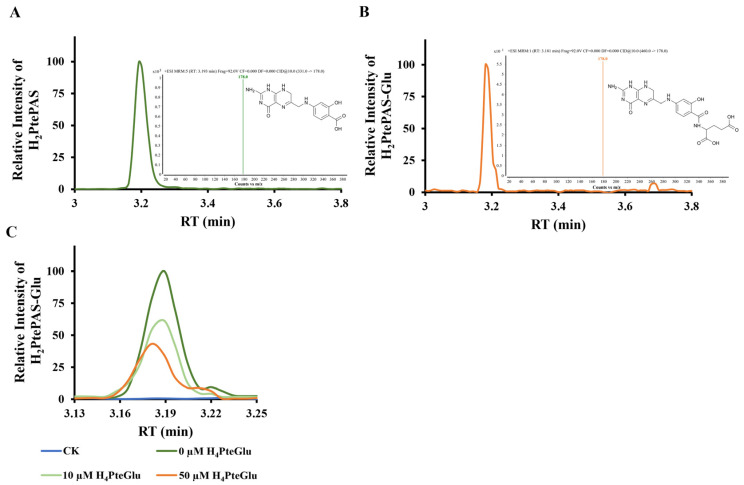
H_4_PteGlu hinders the activation of PAS. (**A**) H_2_PtePAS was identified based on HPLC-MS/MS. Retention time 3.193 min, ion channel 331.0 -> 178.0. (**B**) H_2_PtePAS-Glu was identified based on HPLC-MS/MS. Retention time 3.181 min, ion channel 460.0 -> 178.0. (**C**) Extracted ion chromatograms of H_2_PtePAS-Glu showing H_4_PteGlu reduced the catalytic efficiency of FolC on H_2_PtePAS.

**Table 1 antibiotics-13-00013-t001:** *thyA* deletion confers PAS resistance in *M. tuberculosis* H37Ra.

Strains	MIC to PAS (μg mL^−1^)
H37Ra pMV261	0.04
H37Ra Δ*thyA* pMV261	10.24
H37Ra Δ*thyA* pMV261::*thyA*	0.32
H37Ra Δ*thyA* pMV261::*thyX*	10.24
H37Ra pMV261::*thyA*	0.32
H37Ra pMV261::*thyX*	0.32

**Table 2 antibiotics-13-00013-t002:** Over-expression of *folC* gene reverses the PAS resistance phenotype.

Strains	MIC to PAS (μg mL^−1^)
H37Ra pMV261	0.04
H37Ra pMV261::*folP1*	0.01
H37Ra pMV261::*folC*	0.02
H37Ra pMV261::*dfrA*	81.92
H37Ra Δ*thyA* pMV261	10.24
H37Ra Δ*thyA* pMV261::*folP1*	2.56
H37Ra Δ*thyA* pMV261::*folC*	0.64
H37Ra Δ*thyA* pMV261::*dfrA*	10.24
F461 *	500
F461 pMV261::*folC*	50

* Clinically isolated PAS resistant strain with *thyA* R235P mutation.

## Data Availability

Data will be made available on request.

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
