# Peer review of "Competition between H4PteGlu and H2PtePAS Confers para-Aminosalicylic Acid Resistance in Mycobacterium tuberculosis"

_antibiotics, 2023, doi:10.3390/antibiotics13010013_

Round 1

Reviewer 1 Report

Comments and Suggestions for Authors

The overall composition of the manuscript is good. The paper is scientifically and methodologically accurate. It is interesting work. This manuscript will find interest in many readers.

My recommendation is 'Minor Revision'. More detailed comments are given below.

 General:

1) The abstract should be improved. The way it is presented is confusing.

2) It is requested that the introduction of the main document be improved.

3) The authors need to reinforce their focus on why they carry out this work. The way as presented is weak. Besides, it is suggested that the techniques used to responds to the question of this study will be indicated before proceeding to the results.

Minor comments:

1) To better understand of readers, please define the abbreviation when it first appears. Verify this information in all the main text. The most of the definitions appear in the Materials and Methods section. For example, in the abstract only H4PteGlu is defined; while H2PtePAS-Glu and FolC are not defined.

2) Figure 1. The mechanism of PAS action is missing in the main document. Foe example, 5.10-m-H4PteGlu remains to be described. Please verify that everything is described and defined.

3) Lines 90-91. Please verify this sentence “We noticed that, overexpression of thyA and thyX both caused an 8 times increase in PAS MIC (Table 1).” In the results only thyA showed a increase in PAS -MIC.

 4) Figure 2. Authors should indicate the loading control used in the WB. This must be indicated on the Figure.

 5) Line 274. Please define “PAS MICs” in order of appearance.

 6) Line 274. What´s mean “determination of PAS MICs”. The MIC was determined of the extraction of genomic DNA? Please verify this information.

 7) Line 263. Please homogenize mL “25 μg ml-1 and 100 263 μg mL-1 “ Please verify this information.in all the main doucment and figures.

 8) 4.7. In Vitro Enzymatic Activity Assays: Please explain in the main document why you used H4PteGlu as substrates if it is not catalyzed by FolC?.

 9) The conclusions must be improved. Does not reflect the art of the article and are not coincident with the showed in the abstract section (lines 245-249).

Comments on the Quality of English Language

None

Author Response

The overall composition of the manuscript is good. The paper is scientifically and methodologically accurate. It is interesting work. This manuscript will find interest in many readers.

My recommendation is 'Minor Revision'. More detailed comments are given below.

General:

1)The abstract should be improved. The way it is presented is confusing.

Response: Thank you for your suggestion. We have modified it as your advice.

2)It is requested that the introduction of the main document be improved.

Response: Thank you for your suggestion. We have modified it as your advice.

3)The authors need to reinforce their focus on why they carry out this work. The way as presented is weak. Besides, it is suggested that the techniques used to responds to the question of this study will be indicated before proceeding to the results.

Response: Thank you for your suggestion. We have modified it as your advice.

Minor comments:

1) To better understand of readers, please define the abbreviation when it first appears. Verify this information in all the main text. The most of the definitions appear in the Materials and Methods section. For example, in the abstract only H4PteGlu is defined; while H2PtePAS-Glu and FolC are not defined.

Response: Thank you for your suggestion. We have defined all abbreviation in the main text.

2) Figure 1. The mechanism of PAS action is missing in the main document. For example, 5.10-m-H4PteGlu remains to be described. Please verify that everything is described and defined.

Response: Thank you for your suggestion. The mechanism of PAS action was described in Line 59-63 of the revised MS. 5, 10-m-H4PteGlu was described in Line 155-158. We have added the contents for 5, 10-m-H4PteGlu into the introduction section of the modified MS as your advice (Line 53-56).

3) Lines 90-91. Please verify this sentence “We noticed that, overexpression of thyA and thyX both caused an 8 times increase in PAS MIC (Table 1).” In the results only thyA showed an increase in PAS -MIC.

Response: Thank you for the reminder. We actually showed that overexpression of thyA and thyX both caused an 8 times increase in PAS MIC. In Table 1, we show that the MIC to PAS of H37Ra pMV261 is 0.04, and MIC to PAS of H37Ra pMV261::thyA or H37Ra pMV261::thyX is 0.32.

4) Figure 2. Authors should indicate the loading control used in the WB. This must be indicated on the Figure.

Response: Thank you for the reminder. Actually, there is no commercial loading control used in WB for M. tuberculosis. Thus, the total protein was normalized for loading by NanoDrop2000, and the consistency of the loadings was determined by SDS-PAGE (see the upper part of Fig. 2A). This method is recognized for WB analysis in bacteria. (DOI: 10.1016/j.jare.2021.11.015; DOI: 10.1016/j.ebiom.2022.103943). Previously, we published a work “Deletion of sigB Causes Increased Sensitivity to para-Aminosalicylic Acid and Sulfamethoxazole in Mycobacterium tuberculosis”, which used the same method (DOI: 10.1128/AAC.00551-17). And, we have added the contents for WB assay into the Materials and Methods section of the modified MS as your advice (Line 348).

5) Line 274. Please define “PAS MICs” in order of appearance.

Response: Thank you for your suggestion. We have modified it as your advice.

6) Line 274. What´s mean “determination of PAS MICs”. The MIC was determined of the extraction of genomic DNA? Please verify this information.

Response: Thank you for the reminder. “Single colonies were purified and liquid cultures were prepared for the extraction of genomic DNA and determination of PAS MICs, separately”. We have modified it (Line 308-310 of the revised MS).

7) Line 263. Please homogenize mL “25 μg ml-1 and 100 263 μg mL-1”. Please verify this information.in all the main document and figures.

Response: Thank you for your suggestion. We have modified it as your advice.

8) 4.7. In Vitro Enzymatic Activity Assays: Please explain in the main document why you used H4PteGlu as substrates if it is not catalyzed by FolC?

Response: Thank you for the reminder. FolC was demonstrated to be a bifunctional enzyme in E. col which not only converts H2Pte into H2PteGlu, but also adds glutamic acid tail to H4PteGlu (Line 169-170). In this study, we furtherly demonstrated that H4PteGlu could be catalyzed by FolC.

9) The conclusions must be improved. Does not reflect the art of the article and are not coincident with the showed in the abstract section (lines 245-249).

Response: Thank you for your suggestion. We have modified it as your advice (Line 276-277, Line 280-283 of the revised MS).

Reviewer 2 Report

Comments and Suggestions for Authors

Dear Authors,

The manuscript entitled “Competition between H4PteGlu and H2PtePAS Confers para-Aminosalicylic Acid Resistance in Mycobacterium tuberculosis”, is an interesting study of the association between the misfunction of thyA (deletion) in M. tuberculosis and increasing the content of tetrahydrofolate (H4PteGlu) in bacterial cells and consequent resistance to PAS (anti-tuberculosis drug).

Please find my comments as follows:

Line 21: This sentence should be revised. You previously stated that deletion of ThyA leads to increasing H4PteGlu. But in line 21 you described "loss of function mutations in thyA led to increased H4PteGlu".

Line 22: define H2PtePAS and any other acronyms at their first use

Line 38: You may explain something about the efforts in using phages as an individual or supplementary therapy to treat mycobacterial infections caused by pathogenic mycobacteria such as Mycobacterium tuberculosis. You may cite the following article:

Hosseiniporgham, S.; Sechi, L.A. A Review on Mycobacteriophages: From Classification to Applications. Pathogens 2022, 11, 777. https://doi.org/10.3390/pathogens11070777

Line 39: as a first line, not an first line

Line 40: rewrite this sentence:  It was about 70 years after PAS was first used in clinic that its mechanism of action has been gradually revealed. You may write: the mechanism of action of PAS was gradually discovered over 70 years after its first use in clinics.

Comments on the Quality of English Language

Minor editing of English language required.

Author Response

The manuscript entitled “Competition between H4PteGlu and H2PtePAS Confers para-Aminosalicylic Acid Resistance in Mycobacterium tuberculosis”, is an interesting study of the association between the misfunction of thyA (deletion) in M. tuberculosis and increasing the content of tetrahydrofolate (H4PteGlu) in bacterial cells and consequent resistance to PAS (anti-tuberculosis drug).

Please find my comments as follows:

Line 21: This sentence should be revised. You previously stated that deletion of ThyA leads to increasing H4PteGlu. But in line 21 you described "loss of function mutations in thyA led to increased H4PteGlu".

Response: Thank you for your suggestion. Gene deletion represents a complete loss of biological function of ThyA. Those thyA mutations related to PAS resistance also lead to significant decrease (though not complete loss) of the function (enzymatic activity) of ThyA. Complete or partial loss of ThyA function can lead to increased H4PteGlu content in the bacterial cells.

Line 22: define H2PtePAS and any other acronyms at their first use

Response: Thank you for your suggestion. We have defined all acronyms at their first use.

Line 38: You may explain something about the efforts in using phages as an individual or supplementary therapy to treat mycobacterial infections caused by pathogenic mycobacteria such as Mycobacterium tuberculosis. You may cite the following article: Hosseiniporgham, S.; Sechi, L.A. A Review on Mycobacteriophages: From Classification to Applications. Pathogens 2022, 11, 777. https://doi.org/10.3390/pathogens11070777

Response: Thank you for your suggestion. We have modified it as your advice (Line 44-46 of the revised MS).

Line 39: as a first line, not an first line

Response: Thank you for your suggestion. We have modified it as your advice (Line 57 of the revised MS).

Line 40: rewrite this sentence: It was about 70 years after PAS was first used in clinic that its mechanism of action has been gradually revealed. You may write: the mechanism of action of PAS was gradually discovered over 70 years after its first use in clinics.

Response: Thank you for your suggestion. We have modified it as your advice (Line 58-59 of the revised MS).